# Preconcentration and Removal of Pb(II) Ions from Aqueous Solutions Using Graphene-Based Nanomaterials

**DOI:** 10.3390/ma16031078

**Published:** 2023-01-26

**Authors:** Krystyna Pyrzynska

**Affiliations:** Department of Chemistry, University of Warsaw, Pasteura 1, 02-093 Warsaw, Poland; kryspyrz@chem.uw.edu.pl

**Keywords:** graphene-based nanomaterials, Pb(II) ions, adsorption, aqueous solutions

## Abstract

Direct determination of lead trace concentration in the presence of relatively complex matrices is often a problem. Thus, its preconcentration and separation are necessary in the analytical procedures. Graphene-based nanomaterials have attracted significant interest as potential adsorbents for Pb(II) preconcentration and removal due to their high specific surface area, exceptional porosities, numerous adsorption sites and functionalization ease. Particularly, incorporation of magnetic particles with graphene adsorbents offers an effective approach to overcome the separation problems after a lead enrichment step. This paper summarizes the developments in the applications of graphene-based adsorbents in conventional solid-phase extraction column packing and its alternative approaches in the past 5 years.

## 1. Introduction

Among the heavy metals, lead is extremely toxicant, affecting multiple body systems, and is particularly harmful to young children. Exposure to lead can result in significant health issues, such as damage to brain, kidney, liver, bones, blood and the nervous system [1,2]. The main contamination sources of this element are anthropogenic, such as chemical and battery manufacturing, plastic and printing industries, smelting and mining operations. Due to its non-biodegradable nature and continuous use, lead content accumulates in the environment. Keeping in view the severe toxicity of lead ions, the Environmental Protection Agency (EPA) has established 1.0 mg/L as the maximum permissible concentration of Pb in industrial wastewater and 0.015 mg/L in drinking water, while the World Health Organization (WHO) has set a maximum guideline value of 0.01 mg/L. The European Commission (EC) has proposed lead limits down to 0.005 mg/L [3]. Thus, the development of new analytical procedures for the accurate and sensitive determination of lead is of great importance.

Despite the high sensitivity provided by the atomic and mass spectrometric techniques in trace analysis, a preconcentration step is very often needed to improve their limits of detection further. Additionally, matrix components which cause spectral interference can be removed during this step. Solid phase extraction (SPE) is commonly used as a preconcentration technique owing to its simple procedure, high preconcentration factor and minimal regeneration costs [4,5]. The application of alternative approaches to conventional SPE column-packing such as solid phase microextraction (SPME) [6,7] and dispersive micro solid phase extraction (DMSPE) [8,9] have been also evaluated with carbon-derived materials in the sample preparation step. In recent years, analyte-loaded nanoparticles with magnetic properties were introduced that can be easily isolated from a sample solution by applying an external magnetic field [10,11].

Industrial effluents can contain dissolved lead as well as its various compounds, such as lead salts, oxides, and sulfides. Lead accumulation over time can cause severe ecological problems in water reservoirs due to their toxicity and biocumulation in food chains [12]. It is also a potential health hazard for humans and plants as lead has the tendency to be absorbed through the skin, respiratory and digestive systems. Thus, the need for efficient methods of removing lead ions from wastewater before release into the environment is a matter of concern [13,14,15]. Various treatment technologies have been employed for the removal of lead, such as chemical precipitation, adsorption, ion exchange, chemical oxidation, membrane-based filtration, electrochemical treatment. The selection of the most suitable treatment for lead-contaminated wastewater depends on its initial concentration, pH, environmental impacts as well as economic parameters compared to other technologies. Generally, most of the recent studies have focused on adsorption due to easy operation, high sorption capacity and improved selectivity for specific metal ions. Other techniques have some limitations such as generation of a large amount of sludge, low efficiency, high energy consumption and costly disposal.

Over the last decade, various nanomaterials have been widely applied as powerful adsorbents for trace metal preconcentration and removal of environmental contaminants as they exhibit large surface area, fast adsorption capability and easy functionalization or coatings [16,17,18,19,20]. Among them, graphene derivatives such as graphene oxide and reduced graphene oxide have been integrated in various sample preparation steps, improving the detection sensitivity and selectivity [21,22,23,24]. As these nanomaterials utilise mostly one type of interaction, nanocomposites can combine the properties of carbon materials with metal and metal oxide nanoparticles or chelating polymers [25,26,27]. This allows for further improvement in efficiency of extraction as a result of the synergistic effects resulting from interactions between different materials. Nanocomposities exhibit intrinsic surface reactivity and can strongly chemisorb several substances, with applications in a variety of fields. Their properties can be easily modulated by controlling the methods of their production and additionally the versatile synergistic interactions.

This review presents the last five years of achievement in the use of graphene-based nanomaterials in solid phase extraction for preconcentration of lead ions in analytical processes as well as its removal from aqueous solutions such as sewage. Different works that provide interesting results for this application are summarized. Interested readers could find more details concerning earlier contributions in the review papers [27,28,29,30,31].

## 2. Graphene and Its Derivatives

In recent years, carbon-based nanomaterials, such as graphene (G), graphene oxide (GO) and reduced graphene oxide (rGO), have become one of the research hotspots. They exhibit high surface area, unique structural regularity, chemical inertness, electrical conductivity, thermal and mechanical stability. Graphene is considered the fundamental structure of all carbon allotropes and much research has been done in the field of analytical extraction processes involving graphene and its derivatives as the main material [30,32]. Graphene is a single-atom-thick sheet of sp^2^-hybridized carbon atoms arranged in a planar honeycomb structure. It can be stacked to form graphite and rolled to form carbon nanotubes (Figure 1) [32]. According to Siegel, the nanomaterials can be classified into four categories depending on the dimension in which the size effect on the resultant property becomes apparent [33]. Zero dimensional (0D) nanomaterials (fullerenes and quantum dots) have all the three dimensions less than 100 nm. Carbon nanotubes belong to a one dimensional (1D) category where only one dimension is larger than 100 nm. Graphene is an example of a two dimensional (2D) nanomaterial with two dimensions greater than 100 nm, while graphite (no dimensions are greater than 100 nm) belongs to three dimensional (3D) nanostructures.

However, the agglomeration of graphene sheets due to the strong van der Waals interactions causes a great loss of accessible surface area for the adsorbates. It can be avoided selecting suitable dispersion methods and functionalizing the fillers [34].

Graphene oxide is produced by oxidation of graphite, which leads to increased spacing between the layers. The classic chemical oxidation methods for the preparation of GO utilize strong oxidants such as H_2_SO_4_, KMnO_4_ and NaNO_3_ [35]. Several modifications based on the Hummers method include preoxidation treatment, changing the oxidation intercalation agent, and electrochemically, ultrasonic or microwave assisted methods. A detailed review of GO preparation methods by oxidation of graphite is given by Singh et al. [36].

GO exhibits good hydrophilicity and dispersion in water unlike graphene and graphite powder because of the presence of oxygen containing functional groups, such as –OH and –COOH attached to graphene layers. These functional groups present on the edges and the basal plane of GO play an important role in the the removal processes of heavy metal ions. GO can form magnetic nanocomposites through electrostatic interactions between the negatively charged GO nanosheets and the positively charged surface of Fe_3_O_4_ nanoparticles, combining the high adsorption properties of GO and the convenience of magnetic separation of Fe_3_O_4_ nanoparticles [22]. Furthermore, graphene and GO can be modified or functionalized with the appropriate groups, which facilitates the preparation of hybrid nanoparticles. [23,24,27,32].

Reduced graphene oxide is the form of GO that is processed by chemical, thermal and electrochemical methods [37]. After reduction, the interlayer distance of rGO is decreased due to the partial deoxygenation as well as agglomeration of reduced graphene oxide sheets. The residues of polar groups can improve rGO wettability and it behaves like a hydrophilic-hydrophobic balanced sorbent. The properties of rGO can vary depending on the method of preparation. The scheme of a typical synthesis procedure for graphene derivatives from graphite as a starting material is presented in Figure 2 [38].

## 3. Adsorption Parameters

The important parameters which affect the efficiency of adsorption include pH value of the solution, surface charge of sorbent and its amount, extraction time, temperature as well as concentration and volume of eluent. In acidic medium, the functional groups are protonated and competition occurs in adsorption between H^+^ and Pb(II) ions. With increasing pH value, sorption of metal ions is enhanced. At pH < 6 the predominant lead species is Pb^2+^, while in the range of pH 7–10 Pb(OH)^+^ and Pb(OH)_2_ mainly occur [30]. Thus, in this pH range of solution, besides sorption on the nanoparticles, Pb(OH)_2_ may have a significant participation. On the other hand, as the pH of a solution increases, more hydroxyl and carboxylic acid groups are ionised; thus, the surface charge of carbon-based materials becomes more negative. Li et al. [39] found that both GO and rGO are negatively charged over a very wide pH range but the magnitude of rGO zeta potential was lower than that of GO at the same pH. For this reason the proposed pH for sorption of Pb(II) was mainly in the range of 5–7. Commonly, the efficiency of adsorption increases when the sorbent amount and extraction time increase and equilibrium can be achieved.

Sometimes the presence of variety of metal ions, which are usually found in natural samples, can have a competitive effect on the sorption process of Pb(II) and cause a decrease in its adsorption efficiency. For this reason, high adsorbent capacity as well as selectivity is beneficial. Competitive adsorption experiments using GO showed that the affinity of divalent metal ions decreased in the order: Pb(II) > Cu(II) ≫ Cd(II) > Zn(II) [40]. The dispersibility of GO in water changes remarkably after adsorption of metal ions and the tendency to agglomerate and precipitate was observed.

Usually, the traditional “one variable on time” approach was used to find optimum sample pH, sorbent amount, extraction time and volume of eluent for a Pb(II) extraction process. The chemometric techniques based on factorial designs and response surface methodology were additionally applied in order to identify the relationship between independent variables. As the example of such methodology, Figure 3A shows the response surface plots representing the interaction of the main parameters for Pb(II) removal to find their best combination ensuring the highest process efficiency using EDTA grafted GO and magnetic chitosan [41]. As can be seen, removal efficiency strongly depends on the pH value as this parameter affects the surface charge of the used nanocomposite and subsequently the electrostatic interactions between Pb(II) ions and adsorbent. Moreover, at high levels of both adsorbent and Pb(II), maximum removal efficiency can be achieved. The relative contribution of these parameters decreases in the order: Pb(II) concentration > adsorbent dose > pH > adsorbent dose/Pb(II) concentration ratio (Figure 3B). According to the adaptive neuro-fuzzy inference (ANFI) system and desirability function, an adsorbent amount of 9.75 mg, lead concentration of 22.35 mg/L, pH of 8.38, temperature of 51.89 °C and 22.20 min contact time should provide 90.32% of Pb(II) removal. The values for a genetic algorithm approach were found to be 9.50 mg, 24.00 mg/L, 8.00, 49.32 °C, and 22.20 min, respectively, at which 93.98% removal efficiency could be attained [41].

## 4. Isotherm, Thermodynamic and Kinetic Studies

Adsorption capacity on the studied graphene-based nanoparticles was increased by increasing the equilibrium Pb(II) concentration until saturation of the adsorbent. Mostly Freundlich and Langmuir isotherm models were used to explain the mechanism of Pb(II) uptake, the nature of active sites on their surface and to evaluate the maximum adsorption capacity values. The Freundlich model mainly describes a multilayer and heterogeneous process, while the Langmuir model specifies a monolayer adsorption process on a homogeneous surface with all the adsorption sites of equal affinity. The other employed isotherm models were Dubin–Redushkewich and Temkin models. Considering the obtained regression coefficient (R^2^) of the adsorption isotherms, in most cases the calculated results showed that the Langmuir model gave a better fitting in comparison to to the other three isotherms. The equation for this model is as: *q* = *q*_m_ • *K*_L_
*c*_e_/(1 + *K*_L_*c*_e_), where *q* is the amount of metal ions adsorbed per gram of sorbent, *c*_e_ denotes the equilibrium concentration in solution, *q_m_* is the monolayer theoretical saturation capacity and *K_L_* represents the Langmuir constant that relates to the affinity of binding sites. The values of *q_m_* and *K_L_* were calculated from the intercept and slope of the linear plot of 1/*q* against 1/*c*_e_. The adsorption capacity is an important factor because it determines how much sorbent is required for quantitative separation or preconcentration of the analyte from a given solution.

Azan et al. reported that the Langmuir isotherm showed better fitting for removal of Pb(II) using graphene oxide at 20 and 30 °C temperatures, while at 40 °C, the Freundlich isotherm was more appropriate in describing the adsorption mechanism [42]. It is probably related to distortion of surface homogeneity at that temperature, and as a consequence adsorption was not limited to monolayer formation.

Thermodynamic parameters, ΔG°, ΔH° and ΔS°, are indicators of the possible type and mechanism of the adsorption process. The positive value of enthalpy (ΔH°) specifies that the adsorption process is endothermic in nature, while a positive ΔS° value shows an increase in the randomness at the solid-surface interface. Furthermore, the Gibss free energy (ΔG°) can be evaluated according to the equation: ΔG° = ΔH° − TΔS°. The mostly negative reported ΔG° value demonstrated the spontaneous nature of the adsorption process at room temperature. For example, Foroughi et al. reported enthalpy changes (ΔH°) of −21.62 to −26.65 kJ/mol, entropy changes (ΔS°) of 0.1259 kJ/mol·K and Gibbs free energy (ΔG°) of 15.90 kJ/mol for adsorption of Pb(II) onto GO@chitosn@Fe_3_O_4_-EDTA nanocomposite in the temperature range of 298–338 K [41]. These results revealed an endothermic and spontaneous adsorption process.

In order to compare the kinetics of lead ion removal in a quantitative way, the experimental data were fitted to the kinetic models. Pseudo-first and pseudo-second order models are the most often employed approaches. The rate constants k_1_ and k_2_ were evaluated from the most common linearized forms of these models:ln(q_0_ − qt) = ln(q_e_) − k_1_t  pseudo-first order
t/q_t_ = 1/k_2_q_e_^2^ + t/q_e_  pseudo-second order

The vast majority of published papers reported better fitting to pseudo-second-order model, suggesting a chemisorption proces. However, some papers criticize the use of the lineralized form of that model [43,44,45].

For further analysis of the adsorption kinetics, the diffusion model proposed by Weber and Morris is considered. This model states that when the plot q(t) = f(t^0.5^) is linear and passes through the origin (0,0), the intraparticle diffusion is the dominating limiting factor. For the case where the linearity is ensured but the plot does not pass through the origin, the adsorption may be limited by film diffusion. Such conclusions have been drawn studying the kinetics of adsorptive removal of lead from water using graphene oxide [42].

## 5. Preconcentration of Pb(II) for Analytical Applications

The composition of the analyzed sample is often very complex and the concentration of Pb(II) ions is low. The matrix components could cause several interferences during recording of the analytical signals, and thus, obtaining an accurate and reliable result is very difficult. Using a preliminary preconcentration step in the analytical procedure, the interference components are removed with simultaneous improvement of the method sensitivity. Choosing selective sorbent for this purpose requires consideration of the sample matrix and technique for the final detection, while using adequate experimental conditions such as sorbent mass, volume of loading sample and eluent higher enrichment factors is important. The eluent volume has a significant effect on obtaining the highest analytical signal. Low volume would not permit quantitative desorption, while a large volume dilutes the analyte and consequently the value of the enrichment factor (EF, defined as the ratio of sample volume to eluent volume) is decreased.

In comparison to conventional sorbent materials, graphene-based nanoparticles possess great properties, such as high surface area, enhanced mechanical and chemical stability, and possibility of functionalisation (e.g., hydroxylation, carboxylation, amidation, thiolation, silinization and polymer grafting) to increase their selectivity. Thus, they have found several applications in analytical extraction processes [27,46,47]. In the classical SPE technique, GO and rGO can be packed in minicolumns, cartridges and pipette-tips or implemented in polymeric membranes.

Sorption of Pb(II) on an unmodified GO surface mainly involves the electrostatic attraction between the opposite charges and surface complexation. However, its recovery using dispersive solid-phase extraction in the pH range of 4–6 was only about 60% [48]. With rGO, under the same conditions, the metal removal increased to 90%. Graphene oxide, synthesized by the well-known modified Hummer method using graphite from waste dry cell battery, showed in the batch experiments 98.9% removal of Pb from its 10 mg/L solution at pH 4. The maximum adsorption capacity was calculated to be 55.80 mg/g [42], while Guerrero-Fajardo et al. reported impressive adsorption capacity for Pb(II) of 987.3 mg/g for GO prepared from graphite sheets [44]. An even higher value (1119 mg/g) was reported earlier for GO prepared through the oxidation of graphite using potassium dichromate [49].

The addition of magnetic properties to the GO derivatives facilitates their collection from a sample solution. Such a procedure can be successfully used for extraction purposes, even from difficult to handle samples, without high speed centrifugation and filtration. Soylak et al. reported at least 95% recovery of lead ions at pH 6 (from a solution containing 10 µg of Pb in 100 mL) on magnetic graphite oxide at 2 min of time with the vortex mixing [50]. The maximum sorption capacity of 9.9 mg/g was obtained and the limit of detection was 28 µg/L using flame atomic absorption spectrometry (FAAS) for detection. Due to the corrosive power of 3 mol/L HNO_3_ in 10% acetone used as the eluent, that sorbent was replaced after 10 cycles of sorption/desorption processes. With partially reduced graphene oxide-Fe_3_O_4_ composite, prepared through in situ coprecipitation, the equilibrium was reached in 10 min with the sorption of 95.8% and 373.1 mg/g quantity [51]. For protection of the magnetic core against leaching in acidic media and oxidation/dissolution as well as for improving their reusability, the synthesized Fe_3_O_4_@GO nanoparticles were additionally coated with a silica shell structure [52,53]. The ICP MS detection with a silica-coated magnetic GO nanocomposite preconcentration step exhibited the linear range of 0.05–60 µg/L with the limit of detection (LOD) of 3.64 ng/L for Pb(II) analysis in water samples. Some applications with polymers replacing SiO_2_ for coating can be also found in the literature [54,55].

Several compounds, including well-known reagents and the new materials, have been reported for the functionalization of GO surface that enables more interactions between these functional groups and Pb(II). It has been proved several times that bonding different chemical linkers to graphene or its coating improves the sorption efficiency and selectivity. Thus, a key parameter for development of the new and innovative sorbents for selective preconcentration of lead ions is modification of the carbon surface.

The attachment of dimethylglyoxime ligand to GO@Fe_3_O_4_ adsorbent, using 3-chloropropyl-trimethoxy silane (CPTMS) as a spacer, resulted in higher extraction efficiency and selectivity for Pb(II) in comparison to the direct binding of this ligand to the surface of the nanocomposite [56]. Additionally, smaller particle size with higher ratio of the surface to volume were synthesized. Relatively low EF value (23) and sorption capacity (45.05 mg/g) values were obtained with sonication for 9 min contact time. Sorbent was found to be reusable for at least 14 sequential cycles and was applied for quantification of trace amounts of lead in water, hair and nail samples. This method made possible the determination of Pb(II) in the range of 20.0–600 ng/mL with LOD of 7 ng/mL using flame AAS detection and in the range of 0.5–3.0 ng/mL with graphite furnace AAS (LOD = 0.2 ng/mL).

For magnetic graphite oxide adsorbent with attached 8-hydroxyquinoline ligand, a much higher preconcentration factor of 130 was reported for 5 min of extraction and 22 mg of the sorbent amount [57]. Thus, trace amounts of Pb(II) (simultaneously also Cd(II)) from high-volume samples can be quantified. In addition, a very low limit of detection (0.09 ng/mL) allowed the determination of lead in agriculture products (tomato, mushroom, lettuce) with simply FAAS detection.

According to the reports given by Akbarzade et al. [58], under the optimum conditions (pH, 5.0, 1 mg of adsorbent, 50 μL of 0.4 mol/L HCl for elution), a very high enrichment factor of 600 was reached using magnetic reduced graphene oxide modified with 2-(4-pyridylazo)resorcinol (PAR) by the DSPE method. It was applied for selective preconcentration and determination of lead ions in various waters and food samples. Modification of magnetic graphene oxide with 2-mercapto-benzothiazole (2-MBT) [59] and EDTA [41] have also been proposed.

A schematic diagram for the synthesis of Fe_3_O_4_@GO-MBT nanocomposite is presented in Figure 4 [59]. Briefly, GO sheets were firstly synthesis by the Hummer procedure from graphene powder in H_2_SO_4_. Then, in the ice bath, NaNO_3_, KMnO_4_ and H_2_O_2_ were added slowly and stirred for 45 min. GO sheets were washed in HCl solution, centrifuged and dried. Continuing the procedure, the mixture of Fe(II) and Fe(III) salts was added under stirring, alkalized with ammonia aqueous solution and kept at 90 °C for 4 h under the nitrogen atmosphere. The reaction of magnetic graphene oxide with 3-chloropropyltrimethoxysilane (CPTMS) was carried out under nitrogen atmosphere for 12 h. After washing and drying overnight, GO@Fe_3_O_4_@CPTMS nanocomposites were dispersed in N,N′-dimethylformamide containing 2-MBT. At each stage of relatively long synthesis, the FT-IR spectrum, EDS and TGA analyses were applied to confirm the synthesis and properties of the obtained GO@Fe_3_O_4_, CPTMS@GO-Fe_3_O_4_, and GO@Fe_3_O_4_@MBT nanocomposites. The maximum sorption capacity of the final GO@Fe_3_O_4_@MBT nanocomposite was calculated as 179 mg/g in the presence of Cd(II) and Cu(II) ions, for which also significant values were obtained (164 and 156 mg/g, respectively) [54]. It was applied for preconcentration and separation of Pb(II) prior to its determination in different waters and agricultural products using the FAAS detection method with a linear dynamic range of 1–140 ng/mL and LOD of 0.35 ng/mL.

Seidi et al. [60] synthesized magnetic chitosan-salen grafted GO for lead analysis in whole blood using dispersive SPE coupled with FAAS detection. The proposed Schiff base ligand was synthesized using a mixture of salicylaldehyde and 1,3-diamino-2-propanol in a 2:1 molar ratio. The application of chitosan as a cross-linker ensures the stability of the magnetic nanoparticles and also improves dispersibility of the final product in water solutions. Its sorption capacity was 357 mg/g for an extraction time of 33 min. The protective effect of chitosan was also used for preparation of embedded 1,5-diphenylcarbazone grafted magnetic GO with magnetic chitosan [61]. According to the authors, this effect could be attributed to the increasing chitosan polymer rigidity using glutaraldehyde as a the cross-linker reagent. The preconcentration factor, defined as the slope ratio of the calibration curves with and without a preconcentration step, was 13.5 and a maximum adsorption capacity of 58 mg/g was reported for Pb(II). Finally, the proposed method after an enrichment step followed by FAAS detection with LOD of 0.13 ng/mL was applied for trace lead determination in various water samples. However, the direct analysis enabled Pb(II) quantification only in wastewater. For other samples (tap, well water and seawater), recovery studies were performed to investigate their applicability. The obtained results were in the range of 92.3–109% [61].

The novel nanocomposite of GO@Fe_3_O_4_@chitosan-EDTA was synthesized via an amidation reaction between carboxylic groups of EDTA and amino groups of chitosan [41]. The scheme for its synthesis and the proposed mechanism for its interaction with Pb(II) are presented in Figure 5. Central composite design-based models were developed to evaluate the impact of sorption parameters and describe removal of lead ions. Under the optimum conditions (pH 8, adsorbent dose 9.50 mg and sonication time 22.2 min), its maximum adsorption capacity was specified to be 666.7 mg/g. The significance of the main adsorption parameters decreased in the order: Pb(II) concentration (30%), amount of sorbent (21%) > sample pH (12%). It was postulated that the electrostatic interactions between GO and Pb(II), ion exchange between NH_3_^+^ from chitosan and lead ions, chelation between EDTA and Pb(II) or EDTA with chitosan amine groups were involved. The presence of coexisting ions, such as (Na^+^, K^+^, Mg^2+^, Ca^2+^, Cu^2+^, Zn^2+^, Cl^−^, CO_3_^2−^, and SO_4_^2−^) in environmental aqueous samples as well as in wastewaters, affects the sorption of Pb(II) only at above mixture concentration of 10 mg/L. After six adsorption-desorption cycles only about 8% loss occurred in its efficiency.

Similarly to chitosan, other polymers and their mixtures were successfully loaded onto Fe_3_O_4_@GO surface using in situ polymerization, such as polyimide [62], polyaniline-polypyrolle [63] and polypyrrole-polythiophene [64]. Further, magnetic reduced graphene oxide was wrapped with polydopamine [65]. In some cases, Fe_3_O_4_@GO adsorbent was firstly modified with silica nanoparticles, which enable retention of the polymer with hydroxyl groups present on the silica surface through hydrogen bonding and electrostatic interactions [63,64]. SEM and TEM images showed uniform wrapping with the used polymers. Only for Fe_3_O_4_@rGO-polyimide material, an inhomogeneous morphology with the size less than 30 nm was reported [62]. Among these nanocomposites with polymers, Fe_3_O_4_@rGO-polydopamine nanocomposite showed the highest preconcentration factor of 200 for Pb(II), but the lowest adsorption capacity (35.2 mg/g) [65]. It can be reused at least eight times with no analytes carrying over during the magnetic SPE procedure.

Ionic liquids (ILs) have recently been widely used for the development of various types of adsorbents, because they can interact with several molecules through different interactions, such as hydrogen bonding, π-π, electrostatic, dispersive and dipolar interactions [66,67]. Moreover, their properties and adsorptive selectivity can be matched by adjusting the structure of their constituent anion (inorganic or organic) and large asymmetric cation. Fe_3_O_4_@GO nanospheres were dispersed in ionic liquid 1-butyl-3-methylimidazolium tetra- fluoroborate (BmimBF_4_) to form a ferrofluid for extraction of Pb(II) complexed with 1,5-bis(di-2-pyridil) methylene thio-carbonyl hydrazine [68]. The ferrofluid and a sample were kept in contact for 5 min and after separation using the magnet, elution was performed with 1 mL of 5% HNO_3_ solution. This was applied in magnetic DMSPE and graphite furnace AAS for determination of Pb(II) in seawater samples in the concentration range of 0.04–0.025 µg/L with LOD of 0.008 µg/L. Similar Fe_3_O_4_@GO/BmimBF_4_ nanocomposite have been proposed by Rofouei et al. but the Pb(II) ions were earlier complexed with 1-(2-pyridylazo)-2-naphthol [69]. Following magnetic separation, the ions were eluted from the sorbent with acetonitrile. Response was linear in the 1.5 to 100 μg/L metal ion concentration range with inductively coupled plasma optical emission spectroscopy (ICP OES) quantification and LOD of 0.1 μg/L. Standard water samples spiked with lead were analyzed with relative recoveries within the range of 90.5% to 107.5%.

Apart from Fe_2_O_3_ or Fe_3_O_4_, various metal and metal oxide nanoparticles have been decorated on the surface of graphene or graphene oxide to enhance their adsorption properties and to broaden the applications [70]. These nanoparticles act also as a stabilizer against the aggregation of individual graphene sheets, which is caused by strong van der Waals interactions. The ceria nanoparticles were attached on the graphene by using non-ionic surfactant Triton-X100 and the obtained graphene/CeO_2_ nanosheets appear as an attractive composite in sorption of Pb(II) [71]. The combination of dispersive solid phase microextraction with detection using energy-dispersive X-ray fluorescence spectrometry (EDXRF) allows direct analysis on the nanocomposite surface without the need for elution which decreases the whole analysis time and reduces the cost. The sorption process is very fast (5 min) and the maximum adsorption capacity for Pb(II) was calculated as 75.6 mg/g.

In recent years, biosorption processes using various natural materials has attracted attention and several studies have reported the development of aptamer-based biosensors or their combination with different nanomaterials for applications in food and environmental analysis [72]. Shamsipur et al. [73] proposed the new magnetic biosorbent obtained by covalent immobilization of aptamer as affinity probe on a Fe_3_O_4_@GO surface (Figure 6). It was applied for preconcentration of Pb(II) based on hairpin oligonucleotides forming G-quadruplex structure in the presence of target ions. An enrichment factor of 50 was obtained for blood and urine samples using 500 µL of 0.4 mol/L EDTA as an eluent. The proposed method was characterized by high selectivity, a low detection limit of 0.05 µg/L and wide linearity of the calibration curve over the range of 0.3–867 µg/L.

The metal-organic frameworks (MOFs) constructed by metal ions or their clusters and organic ligands through coordination bonds are considered as a favorable platform for adsorption of Pb(II) due to large surface areas, permanent porosity, multi-functionalization, changeable structures and open metal sites [74]. The combination of MOFs with magnetic graphene-based materials improved the overall structural properties and introduced magnetic separation. The nanocomposite with Zr(IV) ions (denoted as UiO66-NH_2_) and Fe_3_O_4_@GO nanoparticles exhibited the maximum adsorption capacity at pH 6 for Pb(II) at 344.8 mg/g with 60 min contact time [75]. This high adsorption was mainly attributed to the complexation with the amino groups from UiO66-NH_2_ and oxygen containing groups (mainly hydroxyl and carboxyl) by GO. That nanocomposite selectively adsorbed Pb(II) in the presence of other metal ions such as Na(I), K(I), Al(III), Ni(II), Cu(II) and Zn(II) with an EF value of 40. It showed satisfactory recovery of lead ions from different environmental samples.

Other interesting adsorptive materials include layered double hydroxides (LDHs), two-dimensional nanostructures that consist of layers of metal oxides with a positive charge and charge balancing hydrated anions between the layers [76]. They have the general formula [M^2+^_1–x_M^3+^_x_(OH)_2_]^x+^[(A^n−^)_x/n_mH_2_O], where M^2+^ and M^3+^ are the divalent and trivalent cations, respectively, while A^n−1^ represents the intercalating anions. Several possible mechanisms are proposed for adsorption of metal ions, such as surface complexation, isomorphic substitution, surface precipitation, and electrostatic interaction and chelation. The combination of LDH with GO can increase the the adsorption capacity and selectivity [77,78,79]. An SPE procedure based on GO/ZnCr LDH was developed for determination of Pb(II) in human hair samples followed by GF AAS determination [78]. Under the optimized extraction conditions (sample pH 6, sorbent amount 10 mg, 0.5 mL of NaNO_3_ eluent volume), a preconcentration factor of 10 was obtained and adsorbent capacity was 16.9 mg/g. It should be mentioned that the removal of Cu(II), Cr(VI) and Fe(III) was similar to Pb(II). However, as was examined, these ions did not disturb Pb(II) sorption due to its higher adsorption capacity of GO/ZnCr LDH.

Recently, GO decorated with fullerenol nanoparticles C_60_(OH)_22_ has been proposed for efficient removal of Pb(II) ions in complex matrix samples [80]. The low solubility of fullerenes in water was overcome by a simple hydroxylation reaction with NaOH in the presence of quaternary ammonium cations and then grafting the obtained fullerenols to the surface of GO. This research showed enormous adsorption of C_60_(OH)_22_ toward Pb(II) ions of 1307 mg/g at pH 5.5 and high resistance to ionic strength up to 1 mol/L. The experments indicated that the main mechanism of adsorption was an inner-sphere model based on surface complexation with different types of coordination of Pb(II) to the oxygen functional groups.

Examples of the recent applications of graphene-based materials for the preconcentration of Pb(II) using the SPE technique are presented in Table 1.

It is noteworthy to add that graphene-based materials are also used in membrane preconcentration techniques for removal of metal ions [81]. GO-based membrane can be stabilized by hydrogen bonding with cellulose as a matrix [82] or intercalation of oxidized carbon nanotubes [83]. The GO/CNTs membranes can be applied both under vigorous shaking and flow conditions. The affinity of metal ions toward such membranes decreases in the order: Pb(II) > Cu(II) > Cd(II) > Zn(II) > Ni(II) > Co(II) [83]. Adsorption is based on chelation via hydroxyl and carboxyl groups on the membrane surface and the maximum adsorption capacity for Pb(II) ions at pH 5 was calculated as 98 mg/g. The membrane may be used at least for 10 adsorption/desorption cycles.

Together with the fast development of the new nanomaterials for preconcentration purposes, great importance is nowadays attached to the quality of the measurement data in analytical methodologies. The validation of the proposed procedures for Pb(II) determination after an enrichment step was reported in terms of sensitivity, linearity and precision in many of the cited papers. Accuracy of the developed methods was assessed by analyzing appropriate certified reference materials.

## 6. Removal of Pb(II) from Wastewaters

The removing of toxic lead ions from industrial and mining waste effluents has become a key concern [14,15,84,85]. Its presence in the environment may cause long-term health risks to humans and ecosystem. Among various treatments for removal of Pb(II), adsorption methods using synthetic and natural adsorbents offer relatively low costs of operation, materials, and waste discharge. Some disadvantages can be limited adsorption capacity of most applied materials and the need for their more frequent replacement. Thus, adsorption is best used for wastes with moderate to low concentrations of lead.

Various parameters such as temperature, contact time, lead ion concentration, the chemical structure of the adsorbent, its amount, and the presence of other pollutants influence the adsorption process for lead removal. However, the type and chemical structure of an adsorbent and its adsorption capacity are the most critical factors in successful performance.

Graphene oxide functionalized with polyethylenimine (GO-BPEI) in the form of highly porous foams has been proposed for removal of Pb(II) from waters at a large scale [86]. A schematic representation of its synthesis is presented in Figure 7. This nanocomposite exhibits an unusually large adsorption capacity as 3390 mg/g at pH 5, but the kinetics of the adsorption process are very slow; adsorption equilibrium was obtained after 400 min. The GO-BPEI foam saturated with lead ions can be regenerated upon treatment with nitric acid. However, a 20% decrease in the adsorption capacity was observed after ten cycles, probably caused by hydrolysis of amide bonds in the adsorbent structure. The performance of the proposed adsorbent has been checked only for tap water with the presence of alkaline metal ions (up to 75 mg/L) as a matrix but not for real wastewater samples.

Piperazine-modified magnetic graphene oxide (Pip@MGO) nanocomposite also exhibits high adsorption capacity for lead ions, taking advantage of the coordinating capability of piperazine for metal ions and also the high surface area of graphene oxide [87]. At pH 6 it is equal to 558.2 mg/g. In order to evaluate the application of the proposed method in removing lead ions from real samples, river and seawater samples as well as petrochemical wastewater were spiked by known concentrations of Pb(II) at 5 and 10 mg/L concentration level and the adsorption process was performed using only 7 mg of Pip@MGO adsorbent. The recoveries were in the range of 93–99% but the removal efficiency of lead ions was reduced to less than 90% after four consecutive adsorption-regeneration cycles.

Nickel ferrite based reduced graphene oxide (NFrGO) nanocomposite was used by Lingamdinne et al. for the removal of Pb(II) ions from aqueous solution [87,88]. For its preparation, hydrazine hydrate was added to the GO dispersion, followed by nickel nitrate and ferric nitrate solution at pH ≥ 12 (at 2:1 molar ratio) in an inert atmosphere. Then the mixture was heated at 120 °C for 5 h. Ferrite’s nanocomposite can be easily separated using an external magnetic field and can be reused several times without any structural changes. Its maximum adsorption capacity derived from Langmuir isotherm model was 121.95 mg/g at room temperature and this value can be maintained for up to four cycles [88]. Unfortunately, NFrGO nanocomposite, due to the small particle size (32.0 ± 2.0 nm), cannot be directly used in continuous column experiments, only in batch treatment mode. The nanocomposite of graphene oxide with manganese ferrite, developed by a one-pot hydrothermal method, also possesses great adsorption properties for the selective Pb(II) ions removal from the aqueous medium [89]. Adsorption equilibrium occurs after 30 min and gives a maximum adsorption capacity of 621.1 mg/g [90].

Amongst the adsorbents that have been developed to solve water pollution, metal-organic frameworks and their composites have drawn great attention due to their high surface area, ordered porosity and adjustable functionalization [74,91]. Various functional groups can be introduced into MOFs using ligand modification or MOF functionalization to obtain their specific chemical and physical properties.

The incorporation of melamine into Zr-MOFs was successfully performed by the oven-promoted method [92]. It has a sphere-like morphology and a diameter of about 50 nm. The adsorption capacity for Pb(II) was increased to 205 mg/g under the conditions of pH 5 and 40 °C in comparison to the unmodified MOFs (122 mg/g). The coordination interaction between the amino groups and lead ions was confirmed as the adsorption mechanism. Lu et al. proposed a hydrothermal method to modify an Fe-based MOF with graphene oxide, forming a sandwich structure [93]. This composition exhibits the adsorption capacity of 128.6 mg/g and an efficient adsorption rate within 15 min for the removal of lead ions. The introduction into the structure of MOFs of other functional groups, such as thiol, hydroxyl, carboxyl and N-containing groups, has been shown to be highly beneficial for effective Pb(II) removal [92]. For example, Zheng et al. prepared an MOF structure with imidazole groups, that are considered as active sites for adsorbing heavy metal ions. The optimum pH for Pb(II) removal was 7 and the maximum adsorption capacity of this MOF adsorbent was 537.6 mg/g [94]. Its adsorption selectivity to remove Pb(II) from solutions rich in other heavy metals was confirmed. The ability of MOF adsorbent to remove Pb(II) confirmed its superior adsorption selectivity. The authors generally examine these nanocomposities under laboratory conditions, underlining only their big potential for the removal of Pb(II) from industrial wastewater. However, there are no results for real sample applications.

## 7. Conclusions and Future Works

Graphene-based nanomaterials, such as graphene oxide and reduced graphene oxide, have attracted significant interest in recent years for preconcentration and removal of heavy metal ions, including toxic Pb(II), and the number of scientific reports on this subject is constantly growing. They exhibit huge surface area, good thermal and mechanical stability, and ease of surface modification. Similar to carbon nanotubes, graphene-based nanoparticles can be modified using polymers, metal oxides as well as doping with heteroatoms. Functionalization with magnetic nanoparticles facilitates the separation of Pb loaded adsorbent from the solution. In the described works, a particular focus has been placed on the factors affecting the adsorption process; factors such as adsorbent dose, pH, initial concentration, and contact time were addressed, which determine the performance of graphene-based materials as useful adsorbents.

However, it should be noticed that while graphene derivatives have advantages such as high adsorption capacity or selectivity, some drawbacks can also be found for their application in lead preconcentration and removal. The synthesis of the proposed adsorbents is often time-consuming due to multi-stage processes and it requires appropriate high skills.

Future research on graphene-based materials for preconcentration purposes in analytical procedures should concentrate on molecularly imprinted polymers (MIP) due to their excellent selectivity and specific separation. Molecular imprinting has been widely recognized as a potential technique for the synthesis of tailor-made recognition materials through the formation of a polymer network around a template molecule. They are designed with a particular cavity of a suitable size and structure to remove a given analyte by chemisorption or physicosorption process [95]. Although covalent interactions used in adsorption processes can enhance their selectivity compared to non-covalent binding, further quantitative elution of an analyte can encounter difficulties. The imprinted film prepared on the surface of GO or rGO can improve the access to the surface binding sites, increase the response kinetics and enable the rebinding and extraction of the template molecules. Moreover, grafting such adsorbents with magnetic particles makes its separation much easier after the preconcentration step.

The application of natural components with several hydroxyl, amine and carboxyl groups, such as chitosan, cellulose, β-cyclodextrin and amino acids, in the green synthesis of functionalized graphite-based nanoparticles makes an attractive material in both an economic and an environmental perspective. The control of solution pH can change the surface charge of these functional groups. Thus, by choosing appropriate conditions, the electrostatic interaction with Pb(II) ions could be enhanced.

Despite the unique properties of MOFs, such as tunable porosity, large surface area and chemically stability, the number of their applications in wastewater remediation is still limited [96]. Therefore, the development of novel families of water-stable MOFs, with potential application in the technological processes, should be a research area of continued interest.

It should be noticed that most research studies only examine generally carbon-based nanomaterials under laboratory conditions, while wastewaters usually contain different pollutants in different combinations. Thus, further studies are required in large-scale practical environments.

## Figures and Tables

**Figure 1 materials-16-01078-f001:**
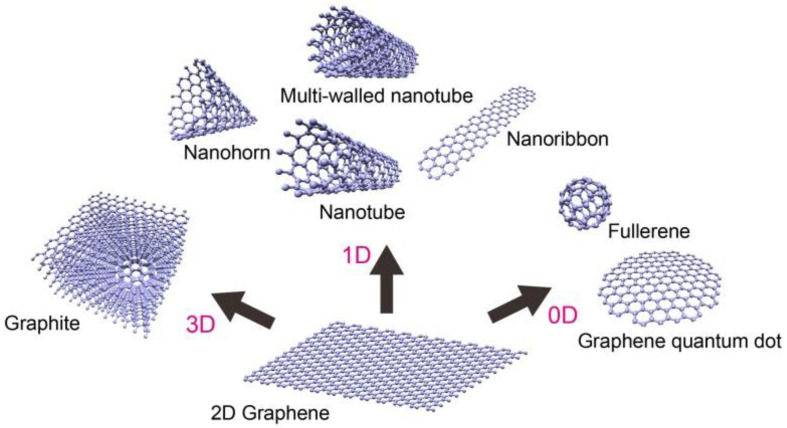
Formation of carbon-based nanomaterials [32].

**Figure 2 materials-16-01078-f002:**
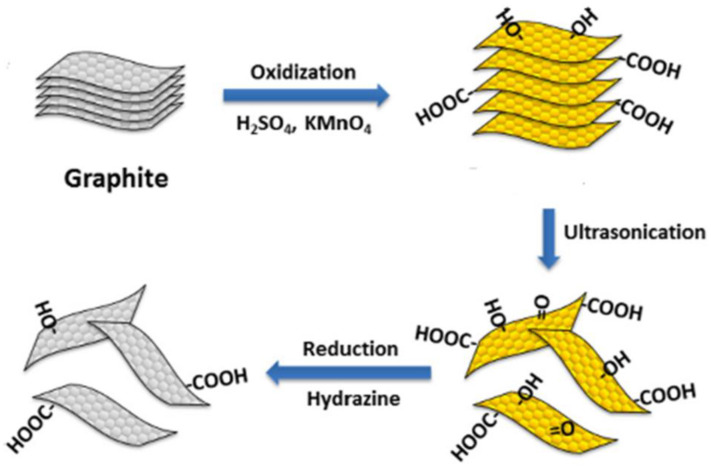
Scheme of a typical procedure for graphene derivatives from graphite as a starting material [33].

**Figure 3 materials-16-01078-f003:**
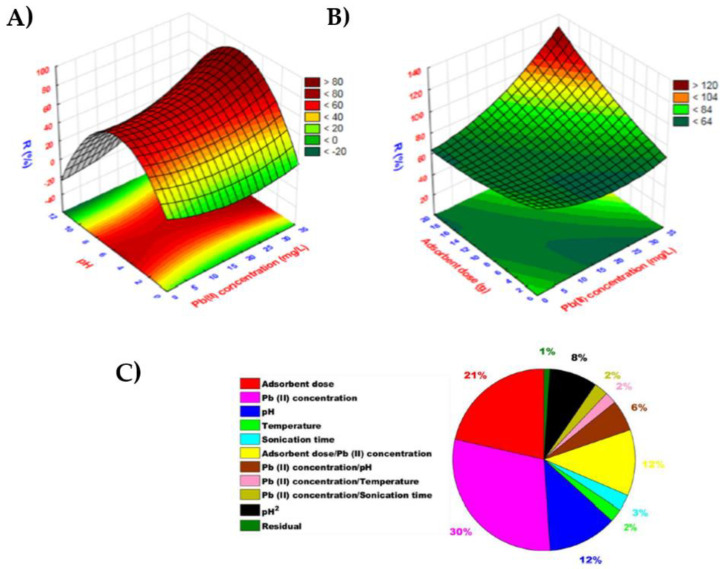
(**A**) three-dimensional response surface plots for the effect of Pb(II) concentration vs. pH; (**B**) the effect of Pb(II) concentration vs. adsorbent dose using GO-chitosan-Fe_3_O_4_-EDTA nanocomposite; (**C**) Relative importance of adsorption parameters for Pb(II) removal using EDTA grafted GO and magnetic chitosan [41].

**Figure 4 materials-16-01078-f004:**
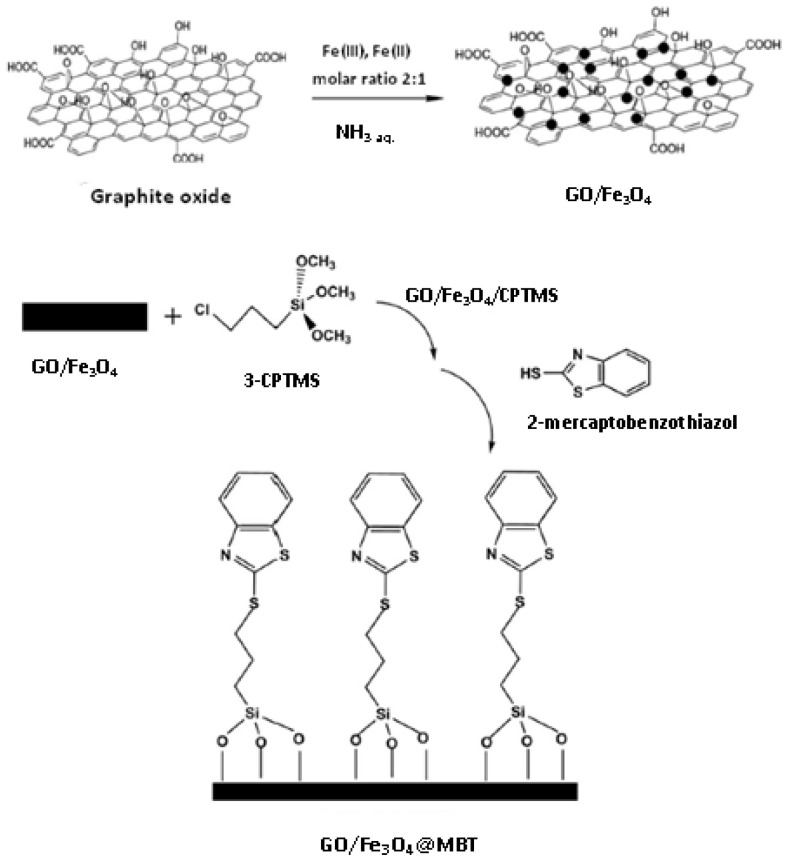
A schematic diagram for the synthesis of 2-mercaptobenzothiazole modified GO/Fe_3_O_4_ nanocomposite [59].

**Figure 5 materials-16-01078-f005:**
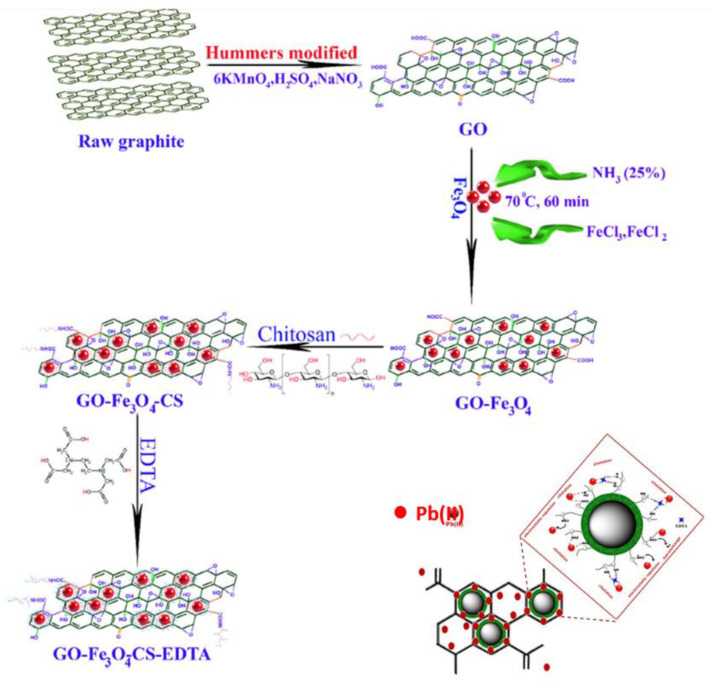
Schematic of the synthesis procedure of GO-Fe_3_O_4_-chitosan-EDTA nanocomposite and the proposed mechanism for its interaction with Pb(II) [41].

**Figure 6 materials-16-01078-f006:**
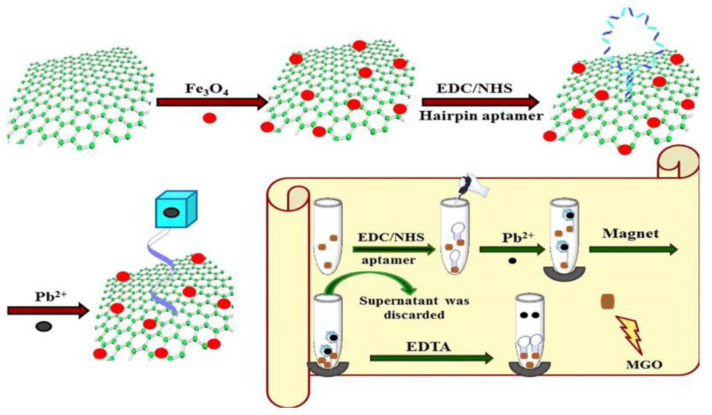
Scheme for synthesis of Fe_3_O_4_@GO nanocomposite with hairpin aptamer and the procedure for preconcentration of Pb(II) using this biosorbent [73].

**Figure 7 materials-16-01078-f007:**
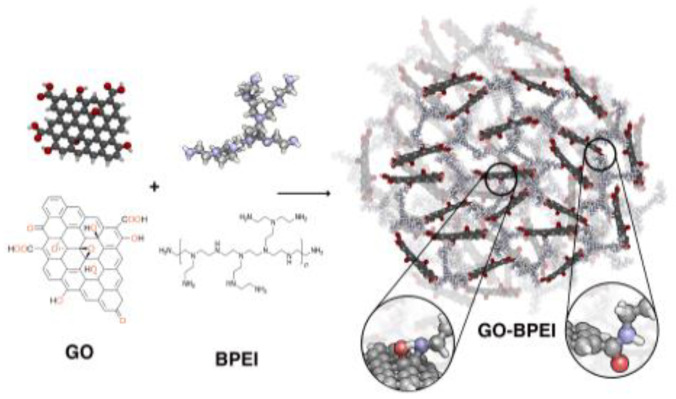
Schematic representation of the GO- branched polyethylenimine synthesis [86].

**Table 1 materials-16-01078-t001:** The examples of the recent applications of graphene-based materials for the preconcentration of Pb(II) using SPE technique.

Sorbent	Sample Matrix	Adsorption		Eluent	EF or PF *	q_max_ (mg/g)	Ref.
		pH	Time (min)				
Fe_3_O_4_@GO@chitosan-Schiff base	Blood	6.3	30	0.8 mol/L HNO_3_	20 *	357	[41]
GO modified with NaOH	Meat	5	5	1.5 mol/L HNO_3_	14 *	25	[48]
Fe_3_O_4_@GO@SiO_2_	Water	5	4	5% HNO_3_	60	168.6	[52]
GO@mesoporous silica	Water, sediment, hair	5	10	0.15 mol/L HNO_3_	15	255	[53]
Fe_3_O_4_@GO@SiO_2_- Dimethyloxime	Water, hair, nail	7.7	9	2.7 mol/L HNO_3_	15.5	45	[56]
Fe_3_O_4_@GO-8HQ	Vegetables, fish, mushrooms	5.5	5	0.74 mol/L HCl	130	150	[57]
Fe_3_O_4_@rGO-PAR	Water, juice, rice	5		0.4 mol/L HCl	600	133	[58]
Fe_3_O_4_@GO-MBT	Water	6	4	0.4 mol/L HCl	400	179	[59]
Fe_3_O_4_@GO@chitosan -DPC	Water	6.2	24	3 mol/L HNO_3_	13.5 *	58	[61]
Fe_3_O_4_/GO@ polyimide	Fish, mollusc tissues	6	15	0.5 mol/L HCl	140	340	[55]
Fe_3_O_4_@GO@SiO_2_ @polyaniline- polypyrrole	Water, rice, milk, wine	5		5% HNO_3_	60	213	[56]
Fe_3_O_4_@GO@SiO_2_@ polypyrrole- polythiophene	Water, apple, tomato	5.8		1 mol/L HNO_3_	36 *	230	[57]
Fe_3_O_4_@GO/BmimBF_4_	Seawater, algae	5	25	0.5 mol/L HCl	200	42.7	[58]
Fe_3_O_4_@GO-hairpin aptamers	Water	with DPTH	5	5% HNO_3_	200	―	[68]
Fe_3_O_4_@GO with aminated MOF	Blood, urine	7	20	0.4 mol/L EDTA	50	―	[73]
GO/ZnCr/LDH	Water	6	60	5% HNO_3_	―	344.8	[75]
GO/C_60_(OH)_2_	Hair	6	―	1 mol/L NaNO_3_, pH 4	10	16.9	[78]
	Highly saline water	5.5	60	―	200	1307	[80]

EF: enrichment factor (the ratio of sample volume to eluent volume); * PF: preconcentration factor (the slope ratio of the calibration curves with and without preconcentration); asterisk: it is connected with PE parameter; q_max_: maximum adsorption capacity; 8HQ: 8-hydroxyquinoline; DPC: 1,5-diphenylcarbazone; MBT: 2-mercaptobenzothiazole; PAR: 2-(4-pyridylazo)resorcinol; BmimBF_4_: 1-butyl-3-methyl imidazolium tetrafluoroborate; DPTH: 1,5-bis(di-2-pyridil)thiocarbonlydrazine; MOF: metal-organic framework; LDH: layered double hydroxide.

## Data Availability

Not applicable.

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
