# Peer review of "Preconcentration and Removal of Pb(II) Ions from Aqueous Solutions Using Graphene-Based Nanomaterials"

_materials, 2023, doi:10.3390/ma16031078_

Round 1

Reviewer 2 Report

The present paper reviews some of technical and scientific aspects of wastewater treatment using nanosorbent produced from graphene derivatives. Different types of grapheme material, effective operating variables and proposed mechanisms involved in lead removal from wastewater have been reviewed. The paper has been focused on a challenging concern in the current environmental issues and thus, it would be worthy for publication after some modifications as follows:

1. Author may rethink about the title. The word “preconcentration” is not a proper word for treatment practices. It is usually used for mineral beneficiation. How do you think about something like: “A critical review on Pb(II) polluted wastewater using graphene-based nanocomposites: technical aspects, mechanisms and future perspective”.

2. Section 3 is very important. However, some response surface plots cannot describe the way operating variables influence the adsorption efficiency. Author is encouraged to prepare a table in which different variables are listed as well as the type of their effects.

3. Section 4 can be divided into two separate sections: application and mechanism(s). Moreover, Table 1 can be interpreted more scientifically as it carries much useful information. You can also classy the graphene-based nanocomposites into suitable groups, e.g. based on nanomaterials used, or number of phases, etc.

4. Please add a section like “Future works” in which you can discuss about the technical shortcomings and scientific gaps as well as works researchers can think about for future research studies.

5. Conclusion should be edited as more representative!

6. Please recheck the literature for some missing works!

Good luck,

Author Response

Please see the atachment

Reviewer 3 Report

The review article " Preconcentration and removal of Pb(II) ions from aqueous solutions using graphene-based nanomaterials" provides a detailed structure of using graphene-based  nanomaterials for the removal of toxic pollutants like Pb were systematically classified, and in-depth discussed.  Below are some of the comments that authors should look into improving the manuscript.

  1. The introduction should be written in a broader perspective and introduce the importance of using graphene based nanomaterials for removal of pollutants.
  2. There are different mechanisms involved in the sorption kinetics, it would be better to give a brief discussion on various models and the importance of sorption behaviour to the models.
  3. The author should give more informations with importance to the conclusions and the future prospectives to improve the efficiency of using GO.
  4. The author has explained about other adsorbates like ionic liquids and magnetic nanoparticles, it would be better to differentiate the importance of choosing GO or combining GO with other adsorbates for the removal of Pb.
  5. It would be better to improve the quality of the figures and some errors in the manuscript.

Round 2

Reviewer 2 Report

The paper is now suggested for publication. 

Author Response

Thank you for the valuable comments regarding my manuscript.

Reviewer 3 Report

There is still some compiling errors in the figures better to correct it

Author Response

Thank you for the valuable comments regarding my manuscript.

Figure 3 was corrected.